# Use of Barbed Sutures for Congenital Diaphragmatic Hernia Repair

**DOI:** 10.3390/children11010035

**Published:** 2023-12-28

**Authors:** Nadine R. Muensterer, Elena Weigl, Anne-Sophie Holler, Christiane Zeller, Beate Häberle, Oliver J. Muensterer

**Affiliations:** Pediatric Surgery, Dr. von Hauner Children’s Hospital, LMU Medical Center, 80337 Munich, Germanyannesophie.holler@med.uni-muenchen.de (A.-S.H.); christiane.zeller@med.uni-muenchen.de (C.Z.); beate.haeberle@med.uni-muenchen.de (B.H.)

**Keywords:** diaphragmatic hernia, congenital malformation, barbed suture, pediatric, thoracoscopy, open repair, patch

## Abstract

Background: Congenital diaphragmatic hernia (CDH) repair can be challenging, particularly when a larger defect is present. Barbed sutures prevent the suture from slipping back after approximation of the tissues. Although introduced almost 2 decades ago, barbed sutures have not been widely used for CDH repair. We report our initial experience and pitfalls. Methods: All patients presenting with CDH from 2021 onward underwent repair using barbed sutures. Demographics, operative parameters, complications, and outcomes were prospectively recorded. Results: A total of 13 patients underwent CDH repair during the study interval (median age 6 days, range 3 days to 5.75 years). Median operative time was 89 min (range 46 to 288 min). Five thoracoscopic and eight open procedures were performed. Severe pulmonary hypertension and ECMO (extracorporeal membrane oxygenation) were considered contraindications for thoracoscopic repair. The included patients were compared to a historic controlled group performed without barbed sutures. The barbed suture facilitated easy and quick closure of the defects in most cases and obviated the need for knot tying. One patient in the thoracoscopic group had a patch placed due to high tension after the barbed sutures tore the diaphragm. At a median follow-up time of 15 months (range 2 to 34 months), one patient had died, and one patient with complete diaphragmatic agenesis was home-ventilated. There were no recurrences. Median operative time trended lower (89 min) than in the historic control group repaired without barbed sutures (119 min, *p* < 0.06) after eliminating outliers with large, complex patch repairs. Conclusions: Barbed sutures simplify congenital diaphragmatic hernia repair regardless of whether a minimal-invasive or open approach is performed. Patch repair is not a contraindication for using barbed sutures. The resulting potential time savings make them particularly useful in patients with cardiac or other severe co-morbidities in which shorter operative times are essential. In cases with high tension, though, the barbs may tear through and produce a “saw” effect on the tissue with subsequent damage.

## 1. Introduction

Congenital diaphragmatic hernia (CDH) is a malformation with a wide spectrum of phenotype, presentation, and outcome [1]. After cardiopulmonary stabilization, surgical closure of the hernia is indicated but can be challenging. This is particularly true in larger defects, as the resulting tension complicates the procedure. Thoracoscopic repair may convey advantages such as shortened postoperative length of stay [2], but it is technically challenging and considered by some to be associated with longer operative times as well as more pronounced acidosis during the procedure [3]. Therefore, technical innovations that simplify the procedure and shorten operative times may help improve outcome.

While there is ample literature on the advantages and disadvantages of thoracoscopic versus open CDH repair [2,3], none of these studies have included the use of barbed sutures. In fact, pediatric surgeons have been very slow to adapt barbed sutures in their armamentarium. This is in contrast to adult surgeons. According to a review of Bishay et al. [3], barbed suture material has been utilized for a wide array of indications, including operations in cosmetic plastic surgery, obstetrics and gynecology, gastrointestinal surgery as well as orthopedic surgery. The authors conclude that barbed sutures present an innovative and effective alternative to conventional surgical sutures [3]. Barbed sutures were also used in pediatric urology. A recent systematic review comparing barbed versus conventional sutures of five studies showed significant shorter operative times when barbed sutures were used [4].

Barbed sutures feature small nicks or hooks along a monofilament thread that prevent it from slipping back after pulling them through the tissues [5]. Therefore, they do not need to be knot tied for secure tissue apposition after typically completing a running stitch. These physical features facilitate the approximation of tissues under a moderate amount of tension, making barbed sutures particularly useful for diaphragmatic hernia repair. Although barbed sutures have been introduced almost two decades ago [6], they have not been widely used for CDH repair. In fact, only one published case series on the topic is available in the medical literature [7]. This study reports our initial experience on CDH repair using barbed sutures and the pitfalls we encountered.

## 2. Materials and Methods

### 2.1. Study Design

All patients presenting with CDH to our department from February 2021 through October 2023 underwent repair using barbed sutures. Demographics, operative parameters, complications, and outcomes were recorded in a registry and later analyzed. The operative time as the primary outcome parameter was statistically compared to a historic control group. All other parameters were defined as secondary measures, which were reported using descriptive statistics in terms of frequency, median and range, and statistically compared.

### 2.2. Patient Selection

All consecutive patients with CDH were included. Co-morbidities and the use of extracorporeal membrane oxygenation (ECMO) were not considered reasons for exclusion. This study also included patients that underwent thoracoscopic hernia repair as well as patients that were operated with an open technique (by laparotomy). In our center, severe pulmonary hypertension and having been treated with ECMO are considered contraindications for a thoracoscopic repair.

### 2.3. Suture Material

The operations were conducted using barbed sutures of 2 different manufacturers (Figure 1). This suture material can be best described as a microfilament suture with barbs or leaflets on its surface allowing for a knotless surgery. The barbs slip through the tissue in an antegrade fashion but hook into the tissue to prevent retrograde pulling through. Thereby, the suture is locked in place. Thus, the need to tie a final knot is eliminated.

### 2.4. Surgical Technique

#### 2.4.1. Open Repair

Open repairs were always completed via a subcostal laparotomy. The infant is placed in a supine position. Subsequently, the abdomen is opened and any abdominal organs that have advanced into the chest cavity are reduced into the abdomen. In the event of a hernia sac, it is resected. After fully exposing and identifying the area of defect, it is categorized according to size (A (small defect) to D (complete absence of the diaphragm), according to Lally et al. [8]). If the size of the defect allows, the defect margins are approximated and closed using barbed sutures. The abdomen is closed in layers.

#### 2.4.2. Thoracoscopic Repair

During a thoracoscopic repair of CDH, the child is positioned in a lateral decubitus position with the affected side’s arm raised up over the shoulders. A 3 mm trocar is advanced into the fourth intercostal space in the mid-axillary line. The capnothorax is insufflated at a pressure of PEEP + 1 mmHg and a flow of 1–3 L/minute. Under vision, two more 3 mm trocars are placed anterior and posterior to the first trocar. Abdominal organs that have herniated into the thoracic cavity are returned to their original location using the intrathoracic pressure and reverse Trendelenburg as well as gentle reduction with a blunt thoracoscopic grasper. If a hernia sac is present, it is pushed back into the abdomen and resected. After the defect margins are identified, the transition of parietal pleura and peritoneum is freshened up using monopolar for improved wound healing, and barbed sutures are utilized to close the diaphragmatic defect (Figure 2; Video in Appendix A). The instruments are retracted and the wounds are closed. Trocars were then retracted. A chest tube may be left postoperatively.

#### 2.4.3. Patch Repair

Some patients required a patch repair in cases when the diaphramatic defect was so large that the margins could not be approximated primarily. There were 2 types of patches available, consisting of polytetrafluoroethylene (PTFE, GoreTex^TM^; W.L. Gore & Associates, Inc, Barcelona, Spain) or collagen (human decellularized collagen, Epiflex^TM^; DIZG. Gemeinnützige GmbH, Berlin, Germany). The patch is generously cut to cover the entire area of the defect. To allow for growth, extra patch material is left on the margins.

Regardless of closure type, a radiograph is performed to verify adequate repair of the defect (Figure 3).

### 2.5. Consent and Ethics

Consent for participation was obtained by all families. Ethical approval for reporting the series was obtained from our institutional ethics board (registration number 23-0881).

### 2.6. Historic Control Group and Statistical Comparison

The last 13 consecutive cases of CDH repair performed in the institution without barbed sutures were compiled and statistically compared with the barbed suture group. Metric parameters were analyzed using the Mann–Whitney U-test, and dichotomous parameters were analyzed using the Chi-square test. A *p* < 0.05 was considered significant, while a *p* < 0.10 was considered a trend. Statistics were calculated using DATAtab statistics.

## 3. Results

### 3.1. Demographic Information of the Patients

Demographic data are presented in the left columns of Table 1. A total of 13 consecutive patients underwent CDH repair in the study interval. Our study comprises nine male and four female patients. The median gestational age at birth was 38 weeks (range 33 to 40 weeks). The median weight at birth was 2960 g (range 1950 to 4270 g) at the time of birth. Ninety-two percent of the patients were diagnosed with co-morbidities, including bilateral vesico-uretero-renaler reflux (VUR), pulmonary hyperplasia, pulmonary hypertension, pulmonary atresia, systemic-pulmonary shunt, and congenital heart defects. One patient presented with volvulus based on malrotation at the time of surgery, one older patient presented with recurrent pneumonia. In ten patients, the liver was located in the chest cavity (“liver-up”). Four patients had been treated with ECMO prior to CDH repair.

### 3.2. Intraoperative Parameters

Table 1 also shows peri- and postoperative outcome parameters of the patients. The median age at the time of operation was 6 days (range 3 days to 5.75 years). The median weight at operation was 3140 g (range 1910 g to 30 kg). Five children underwent thoracoscopic repair, and the other eight underwent open repair via laparotomy. Five patients presented with a right-sided defect. Our study comprises of one type A, seven type B, three type C and two type D CDHs. One patient in the thoracoscopic group and four patients in the laparotomy group had a patch placed. In the thoracoscopic case, the patch was not planned: After attempting a primary repair with barbed sutures, high tension led to a pull-back of the suture against the barbs. These tore the diaphragmatic tissue by a sawing effect of the barbs, representing the only perioperative complication. The median operative time was 89 min (range 46 to 288 min).

### 3.3. Complications and Outcomes

The right-sided columns of Table 1 give an overview of all complications and outcomes. The median postoperative length of stay was 30 days (range 3 to 139 days). At a median follow-up time of 15 months (range 2 to 34 months), one patient had died due to pulmonary hypoplasia and a clotted systemic-pulmonary shunt (mortality 8%), and one patient with complete diaphragmatic agenesis was home ventilated via tracheostomy. There were no recurrences of the CDH, and three patients with larger defects had gastroesophageal reflux requiring proton-pump therapy. One child developed a necrotizing enterocolitis postoperatively that was treated with antibiotics alone (Bell stage IIb). Another was treated with antibiotics due to a central venous line sepsis. One of the infants developed pneumonia seven weeks after the operative procedure, which also required treatment with antibiotics.

### 3.4. Comparison to the Historic Control Group

All parameters of the control group are presented in Table 2. Although the median operative time was lower (89 min) in the barbed group compared to controls (119) minutes, this difference did not reach statistical significance in the Mann–Whitney U-test (*p* = 0.24) when considering all patients. The box plot of operative times in Figure 4 shows two substantial outliers in the barbed suture group. These two patients were associated with complex patch reconstructions of large diaphragmatic defects. There were no comparable cases in the control group. When eliminating these two outliers from the statistical comparison, operative times trended shorter in the barbed suture group (*p* = 0.06). Overall, there were more overall patch repairs in the barbed suture group (*p* = 0.05). Gestational age in the barbed suture group trended lower as well.

## 4. Discussion

Our series represents the largest comprehensive study on consecutive patients who underwent barbed suture repair of congenital diaphragmatic hernia repair across both open and minimal-invasive techniques.

Thus far, most scientific research on CDH repair in infants focuses on the differences between the thoracoscopic and open methods [2,3,9]. Barbed sutures have been used for cruroplasty and hiatal hernia closure in adults [10]. However, they are strikingly absent from the discussion in pediatric surgery. Even research that could potentially be used in the development of future guidelines makes no mention of barbed suture material [1]. Hence, the use of barbed sutures is vastly undervalued and not yet ascertained in the field of pediatric surgery.

There is only one other case series that explores the use of barbed sutures for CDH repair in infants and children, which was published by Lukish et al. [7]. In this case series, eleven patients underwent a minimally invasive diaphragmatic hernia repair (thoracoscopic or laparoscopic) with barbed sutures. In contrast, our patients were operated thoracoscopically or open via laparotomy. Therefore, this is the first series of open CDH repair using barbed sutures. In the previous series [7], 19 percent of patients developed a recurrence, primarily the ones in which absorbable sutures were used. None of our patients had a recurrence. After the initial two patients, we performed all operations using non-absorbable, permanent sutures. In the first two, we intraoperatively resected the hernia sac and freshened up the margins of the defect with the intention that the tissues would heal. Neither one had a recurrence after 30 and 31 months of follow-up.

In the Lukish study [7], none of the other nine patients suffered from complications or mortality. On the other hand, one of our patients died of complications resulting from pulmonary atresia that was treated by cardiac surgery and a systemic pulmonary shunt. Upon autopsy, acute clotting of the shunt was discovered as the probable cause of death. Considering that our cohort was a consecutive, non-selected cohort including some very ill neonates, and out of which four patients were treated with ECMO, mortality actually was quite low.

Lukish et al. [7] emphasized that the “barbed suture is an innovative, safe and time-saving option” for pediatric surgeons. Our findings corroborate this conclusion. They also stress that further prospective analyses with long-term follow-ups are required to confirm the initial results and success with the barbed suture method.

Because of the heterogeneous presentation of CDH, it is difficult to establish an internal control group. However, our study implies that the operative time may potentially be reduced through the use of barbed suture material. We reported a median operative time of 89 min (range 46 to 288 min). In comparison, the median operative time with conventional suture material (not using barbed suture) was reported as 102 min for a thoracoscopic repair and 129 min for an open repair in a recent study [9]. In our analysis, the median operative time for barbed suture CDH repair trended lower compared to the operative time of the historic control group. Because CDH is a rare disease, it was not possible to do a matched-pair analysis, since every patient with CDH is different and has their own characteristics and co-morbidities. Operative time, however, is an objective parameter and therefore chosen as the primary outcome parameter. It is independent of most co-morbidities and mostly influenced by technical factors, such as the type of suture employed. Therefore, the main take-home message of this study is that barbed sutures may shorten operative time of CDH repair.

Regarding the comparison of operative times between groups, the lack of significance is mainly driven by two outliers in the barbed group (patients 6 and 8), who had operations that took 288 and 200 min to complete. Both patients had large (type C and D) defects and required a complex patch repair. The box plot of operative times (Figure 4) illustrates the effect of the outliers. For correct interpretation of the operative times, it is important to note that there was a trend toward more patch repairs and lower gestational age in the barbed suture group, potentially complicating and thereby protracting the operation.

The potential reduction in operative time is most likely attributed to an easier closure without the need for intracorporeal or open knot tying. This method simply allows for a faster and easier closure. This is particularly beneficial if the physiology is tenuous, for example in premature infants with low body weight or patients with complex co-morbidities. We believe that larger patient groups and sample sizes will eventually show superiority of the barbed suture in terms of operative times.

In addition, our findings indicate that the recurrence rate may be reduced through the use of barbed suture material, especially if permanent non-absorbable suture material is used. There were no recurrences in our series of 12 patients. A systematic review on CDH repair using conventional sutures showed a 8.6 percent recurrence rate for thoracoscopic repair and a 1.6% recurrence rate for open repair [11]. Nevertheless, it must be noted that the only other case series on barbed suture CDH repair reports a 19% recurrence rate [7]. Obviously, the small sample size substantially limits adequate judgment at this time.

The median length of hospital stay in our series was not shorter than reported in the literature. While we report a median hospital stay of 30 days (range 3 to 139 days), it was 13 days for thoracoscopic repair and 19 days for open repair in a recent report [9]. However, it must be noted that the vast majority of our patients presented with multiple co-morbidities, many of them complex. Thus, the increased length in hospital stay most likely cannot be attributed to the barbed suture method but rather to a range of health issues. Likewise, the survival rate of 92% in our study compares closely to the survival rate in recent studies [9,11].

Overall, the barbed suture method for CDH repair has few disadvantages. The higher cost associated with barbed suture material (3 to 4-fold of conventional sutures at our institution, around €40 for a barbed suture instead of €12 for a conventional suture) might deter some surgeons from using it, although it is negligible when considering the overall hospital cost of treating a CDH patient. In fact, several studies have shown actually decreased total hospital costs when barbed sutures were used for a variety of different indications [12,13,14].

A definitive pitfall to avoid is the use when tension is too high for the barbed sutures to hold. In that case, the barbed suture slips back through the tissue, and the barbs tear the tissue by a saw effect. This was the case in patient 5. Therefore, barbed sutures are more suitable for cases with low to medium tension. If the defect is too large, a patch repair should be performed. The patch can be fixed to the surrounding edges or the thoracic wall using barbed sutures.

Small bowel obstruction by a lose end of the barbed suture is the most severe complication after use of bowel sutures [15,16,17,18]. The risk can be minimized by leaving a small tail. We usually complete a running barbed suture by making one to two locking loops that prevent the backslip of short suture ends.

Our findings are bound by certain limitations. This study comprises a limited number of cases. The retrospective nature of the study is also a limitation. Additional prospective research with a larger pool of patients and longer follow-ups is warranted for a more precise understanding of the advantages and disadvantages of the barbed suture method in CDH repair.

## 5. Conclusions

Our study indicates that barbed sutures can facilitate the repair of congenital diaphragmatic hernia. They are applicable for both thoracoscopic and open procedures as well as for patch repair. Obviating the need for knot tying may shorten operative times, making barbed sutures particularly useful for CDH repair in infants with tenuous physiology and for minimal-invasive repair. To lower the risk of recurrence, non-absorbable barbed sutures should be used exclusively. More comparative studies are required to objectify the advantages of barbed sutures for CDH repair. Ideally, a randomized controlled trial of barbed versus conventional sutures should be performed with the outcome parameters described in our study. Because CDH is a rare and heterogeneous malformation, a multi-center study approach with a standardized protocol for defect closure should be employed to assure adequate statistical power.

## Figures and Tables

**Figure 1 children-11-00035-f001:**
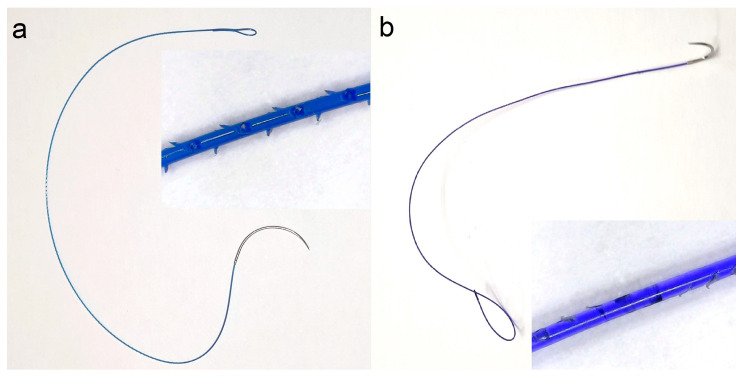
Sutures used in this study. The V-Loc^TM^ (Medtronic GmbH, Meerbusch, Germany) suture (**a**) has small spiral spikes that prevent backslipping of the suture, while the Quill^TM^ (Corza Medical GmbH, Zurich, Switzerland), suture (**b**) features small spiral oblique cut-in leaflets that act in the same manner (see magnified inlays).

**Figure 2 children-11-00035-f002:**
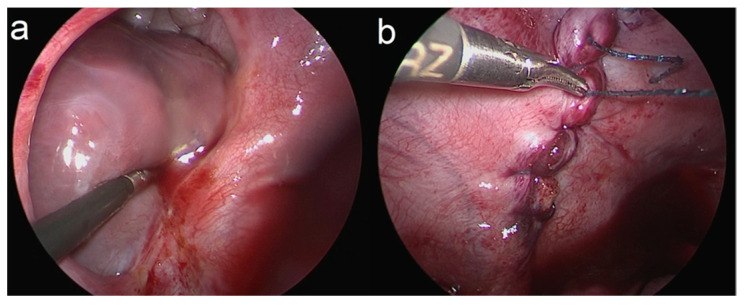
Thoracoscopic closure of a type C defect (**a**) using a barbed suture (**b**).

**Figure 3 children-11-00035-f003:**
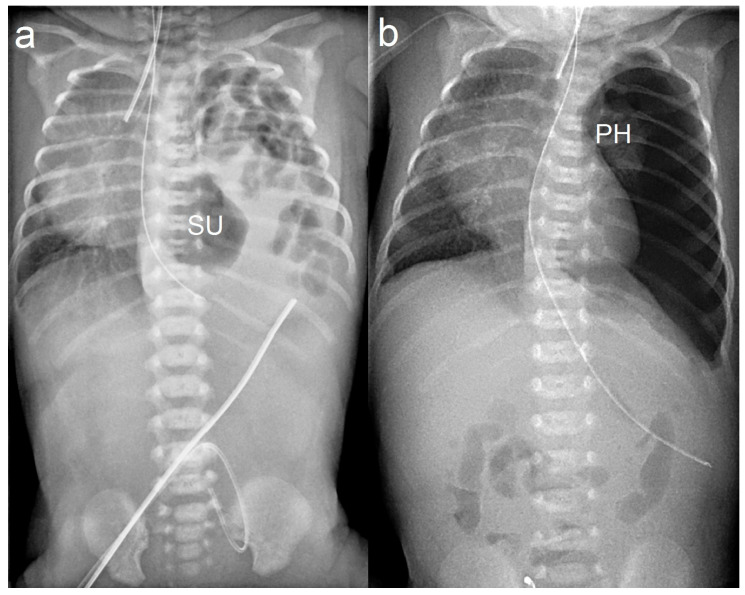
Left-sided CDH with stomach-up (SU) configuration and mediastinal shift toward the right (**a**). After barbed suture open repair via laparotomy, the left diaphragm shows a flat, even contour. Severe pulmonary hypoplasia (PH) is evident (**b**).

**Figure 4 children-11-00035-f004:**
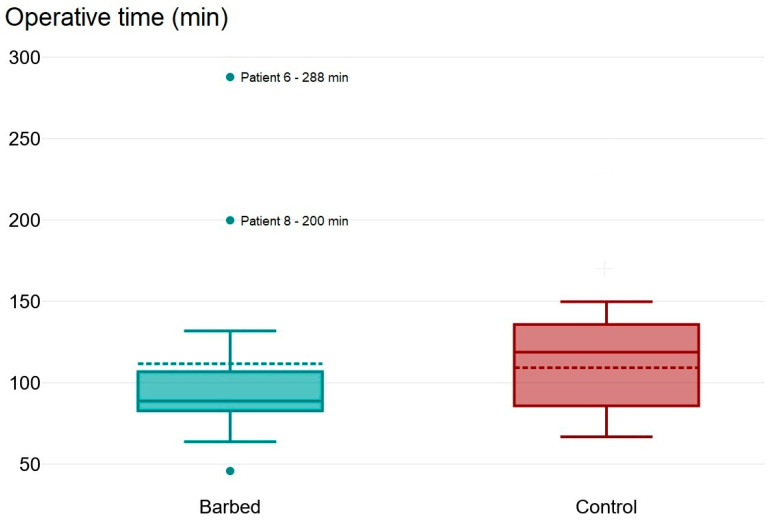
Box plot of operative time in the barbed suture group (**left**, green) compared to controls (**right**, red). Although the Mann–Whitney U-test showed no statistical difference (*p* = 0.24), this lack of significance was mainly driven by 2 severe outliers in the barbed suture group (patients 6 and 8) which had large defects that required a complex patch repair. After eliminating the 2 outliers from the calculation, a trend toward shorter operative times in the barbed suture group was evident (*p* = 0.06).

**Table 1 children-11-00035-t001:** Demographic information, intraoperative findings, postoperative results, complications, and outcomes of patients who underwent congenital diaphragmatic hernia repair using barbed sutures at our institution. The lower 2 rows show descriptive statistics of the variables, shaded in grey.

Patient	Sex	Gestational Age at Birth (wks)	Birth Weight (g)	Co-Morbidities	ECMO	Age at Operation (d)	Weight at Operation (g)	Mode of Operation	Operative Time (min)	Localization of Diaphragmatic Defect and Type	Liver	Patch	Intraoperative Complications	Suture Used	Postoperative Length of Stay (d)	Follow-Up (Months)	Outcome
**1**	M	not available	not available	Colon perforation by chest tube (outside hospital)	No	587	14,800	Lap converted to open	132	Left posterior, Type B	Down	No	None, colon perforation was primarily closed	2-0 V-Loc permanent	3	34	No recurrence, doing well
**2**	M	36	2410	None	No	14	2480	Thoracoscopic	64	Right posterior, Type B	Up	No	None	3-0 V-Loc absorbable	6	32	No recurrence, doing well
**3**	F	33	1950	Bilateral VUR, pulmonary hypoplasia, pulmonary hypertension	No	3	1910	Open abdominal	89	Left posterior, Type B	Up	No	None	3-0 V-Loc absorbable	30	31	No recurrence, doing well
**4**	M	38	2830	Pulmonary atresia, s/p systemic-pulmonary shunt, pulmonary hypoplasia, pulmonary hypertension	No	14	2910	Open abdominal	46	Right posterior, Type B	Up	No	None	3-0 Quill permanent	18	25	Died at age 18 d because of pulmonary hypoplasia, clotted systemic-pulmonary shunt
**5**	M	38	2990	Pulmonary hypertension, PDA	Repair after VA ECMO	6	3440	Open abdominal	98	Left posterior, Type C	Up	Yes, PTFE-collagen Sandwich	None	4-0 V-Loc permanent	90	20	Gastroesophageal reflux, no recurrence, doing well
**6**	M	37	2930	Pulmonary hypertension	No	6	2900	Thoracoscopic	288	Left posterior, Type C	Up	Yes, PTFE-collagen Combination	Sutures pulled back and damaged diaphragmatic tissue because of saw effect; therefore, PTFE patch placed	3-0 V-Loc permanent	38	18	Developed Bell stage IIb necrotizing enterocolitis postop, no recurrence, doing well
**7**	M	40	3420	VSD, ASD, PDA	No	5	3410	Thoracoscopic	77	Left posterior, Type B	Down	No	None	3-0 Quill permanent	43	15	Developed CVL sepsis, treated with antibiotics, no recurrence, doing well
**8**	M	38	3100	Pulmonary hypertension	No	5	3140	Open abdominal	200	Left posterior, Type D	Up	Yes, PTFE	None	3-0 V-Loc permanent	33	14	No recurrence, gastroesophageal reflux, iron-deficiency anemia
**9**	F	38	2260	SGA, pulmonary hypertension	Repair after VA ECMO	8	2200	Open abdominal	83	Right posterior, Type C	Up	Yes, collagen	None	3-0 V-Loc permanent	81	9	Pneumonia at 2 months, no recurrence, slow weight gain, feeding issues
**10**	F	36	2600	Pulmonary hypertension, ASD	Repair after VA ECMO	6	2600	Open abdominal	107	Left, Type D (no apparent diaphragm)	Up	Yes, collagen	None	3-0 Quill permanent	139	7	Failure to wean off ventilator, tracheostomy, home ventilation, no recurrence, gastroesophageal reflux
**11**	M	39	3070	Volvulus based on malrotation, ASD, pulmonary hypertension (NO)	No	6	3140	Thoracoscopic (CDH repair), laparoscopic (Ladd)	89	Left posterior, Type B	Up	No	None	3-0 V-Loc permanent	30	7	No recurrence, doing well
**12**	M	40	4000	Lung hemorrhage, small VSD, adrenal hemorrhage	Repair after VA ECMO	8	4000	Open abdominal	83	Right lateral, Type B	Up	No	Compromised venous return and brief cardiocirculatory arrest upon liver reduction on first attempt. Opened defect further and no issues thereafter	3-0 Quill permanent	17	3	No recurrence, doing well
**13**	F	40	4270	Recurrent pneumonia before diagnosis, no other	No	2099	30,000	Thoracoscopic	98	Right lateral, Type A	Down	No	None	3-0 Quill permanent	6	2	No recurrence, doing well
**Frequency**	4F/9M				4 ECMO			5 Thoracoscopic		7 B, 3 C, 2 D	3 Down, 10 Up	5 Patches					0 Recurrences
**Median [range]**		38 [33–40]	2960 [1950–4270]			6 [3–2099]	3140 [1910–30,000]		89 [46–288]						38 [33–40]	15 [2–34]	

Abbreviations: ASD—atrial septal defect; CDH—congenital diaphragmatic hernia; CVL—central venous line; Lap—laparoscopic; NO—nitric oxide; PDA—patent ductus arteriosus; PTFE—perfluorotetraethylene; SGA—small for gestational age; s/p—status post; VSD—ventricular septal defect; VUR—vesicoureteral reflux; wks—weeks; VA ECMO—venoarterious extracorporeal membrane oxygenation.

**Table 2 children-11-00035-t002:** Overview of the historic control group. *p*-values in the bottom row are calculated from the comparison to the barbed suture group. The lower 3 rows show descriptive and comparative statistics of the variables, shaded in grey.

Patient	Sex	Gestational Age at Birth (wks)	Birth Weight (g)	Co-Morbidities	ECMO	Age at Operation (d)	Weight at Operation (g)	Mode of Operation	Operative Time (min)	Localization of Diaphragmatic Defect and Type	Liver	Patch	Intraoperative Complications	Suture Used	Postoperative Length of Stay (d)	Follow-Up (Months)	Outcome
**1**	M	40	3980	Ectopic left kidney, GERD	No	35	4800	Thoracoscopic	124	Left posterior, Type B	Down	No	None	Silk 2-0, PDS 2-0 pericostal	20	12	No recurrence, doing well
**2**	M	40	3640	Pulmonary hypertension	No	3	3600	Thoracoscopic	150	Left posterior, Type B	Down	No	Lung injury, chest tube	Polyester 3-0, interrupted	20	LTFU	LTFU
**3**	F	40	2995	Pulmonary hypoplasia	No	2	2955	Thoracoscopic	86	Left posterior, Type B	Down	No	None	Polyester 2-0, interrupted	11	7	No recurrence, doing well, PFO
**4**	F	39	2830	Pulmonary hypoplasia, PDA	No	2	2800	Thoracoscopic convert to open abdominal	119	Left posterior, Type B	Down	No	Spleen injury	Polyester 2-0, interrupted	12	LTFO	LTFU
**5**	M	40	3750	Pulmonary hypertension, pulmonary hypoplasia, PFOT	No	3	3700	Thoracoscopic convert to open abdominal	143	Left posterior, Type A	Down	No	None	Polyester 2-0, interrupted	18	9	No recurrence, doing well
**6**	F	40	3720	Pulmonary hypoplasia, pulmonary hypertension, PDA, PFO	No	4	3700	Thoracoscopic	136	Left posterior, Type B	Down	No	None	Polyester 2-0, continuous	4	8	No recurrence, doing well
**7**	M	40	3880	Apnea spells	No	453	8900	Thoracoscopic	67	Left posterior, Type B	Down	No	None	Polyester 2-0, interrupted	20	38	No recurrence, doing well
**8**	F	35	2645	SGA, pulmonary hypoplasia, pulmonary hypertension, PDA, PFO, tracheal stenosis, GERD	No	3	2585	Open abdominal	122	Left, Type D	Up	Yes, PTFE	None	PTFE 3-0, interrupted	131	38	Gastroesophageal reflux, asymptomatic CDH recurrence. underwent slide-plasty of the trachea, gastrostomy, scoliosis
**9**	M	35	2510	SGA, pulmonary hypertension, PDA, PFO, tracheostomy, osseous malformation	No	7	2510	Open abdominal	139	Left, Type D	Up	Yes, PTFE	None	PTFE 3-0, interrupted	173	30	Gastrostomy, 4x CDH recurrence lapse, gastroesophageal reflux, Thal fundoplication, subphrenic abscesses, sepsis, tracheostomy, home ventilation
**10**	F	41	3600	Ladds bands	No	8	3560	Open abdominal	84	Left posterior, Type B	Down	No	None	Polyester 4-0, interrupted	9	6	No recurrence, doing well
**11**	F	38	2540	Pulmonary hypoplasia, pulmonary hypertension, PDA, PFO, thalamic bleed,	Yes	2	2540	Open abdominal	88	Left posterior, Type B	Down	No	None	Polyester 2-0, interrupted	128	4	Tracheostomy and gastrostomy, recurrent bacterial pulmonary infections, died at 4 m of age of respiratory insufficiency
**12**	M	42	3700	Pulmonary hypoplasia, pulmonary hypertension, PDA, PFO, asphyxia	No	3	3700	Open abdominal	89	Left posterior, Type B	Down	No	None	Polyester 2-0., interrupted	22	16	Gastroesophageal reflux, failure to thrive, Thal fundoplication, appendectomy
**13**	M	38	2900	Pulmonary hypoplasia, pulmonary hypertension, PDA, PFO	No	5	2900	Open abdominal	76	Right lateral, Type B	Up	No	None	Polyester 3-0 interrupted	20	LTFU	LTFU
**Frequency**	6F/7M				1 ECMO			7 Thoracoscopic, 2 conversions to open		1 A, 10 B, 2 D	10 Down, 3 Up	2Patches					2 Recurrences
**Median [Range]**		40 [35–42]	3600 [2510–3980]			3 [2–453]	3560 [2510–8900]		119 [67–150]						20 [4–173]	10.5 [4–38]	
**p**	0.85	0.09	0.32	1	0.49	0.03	0.58	0.53	0.24, 0.06 (excluding outliers)	left/right 0.19 A-D 0.77	0.63	0.05	0.51		0.66	0.93	0.24

Abbreviations: wks—weeks; VUR—vesicoureteral reflux; PDA—patent ductus arteriosus; PFO—patent foramen ovale, VSD—ventricular septal defect; ASD—atrial septal defect, SGA—small for gestational age; s/p—status post; NO—nitric oxide; VA ECMO—venoarterious extracorporeal membrane oxygenation, CDH—congenital diaphragmatic hernia, PTFE—perfluorotetraethylene, LTFO—lost to follow up.

## Data Availability

The data presented in this study are available on request from the corresponding author. The data are not publicly available due to privacy and data protection issues of our institution.

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
