# Peer review of "Use of Barbed Sutures for Congenital Diaphragmatic Hernia Repair"

_children, 2023, doi:10.3390/children11010035_

Round 1

Reviewer 1 Report

Comments and Suggestions for Authors

The authors reported their initial experience of repairing congenital diaphragmatic hernias using barbed sutures, including the pitfalls they encountered. They concluded that barbed sutures simplify the repair of congenital diaphragmatic hernias, regardless of whether a minimally invasive or open approach is chosen. Patch repair is not a contraindication to the use of barbed sutures. The resulting potential time savings make it particularly useful in patients with cardiac or other severe comorbidities, in which shorter operative times are essential.

I read the study with great interest. Unfortunately, important methodological issues were raised during the review. My concerns are as follows:

1. Abstract – Authors should indicate the average age of patients in the abstract. In addition, any abbreviation in the abstract should be listed in the full title on the first mention (e.g. ECMO).

2. Introduction – The authors state that barbed sutures have been used for a variety of indications, including cosmetic plastic surgery, obstetrics and gynecology, gastrointestinal surgery, and orthopedic surgery. This is true, but barbed sutures were also used in pediatric urology (e.g. for pyeloplasty). This statement should be added including a recent meta-analysis reference (Anand S, et al. Barbed versus non-barbed suture for pyeloplasty via the minimally invasive approach: A systematic review and meta-analysis. J Laparoendosc Adv Surg Tech A. 2022;32(10):1056-1063. doi: 10.1089/lap.2021.0868).

3. Study design – The authors state that demographics, operative parameters, complications, and outcomes were recorded in a registry and later analyzed. Please indicate for each category the analyzed parameters.

4. Study Design – The main problem with the design of this study is the lack of a comparison group. I would advise the authors to introduce a comparison group (Patients operated on with other types of sutures from previous years). Without a comparison group, this is a descriptive study without clinical or scientific relevance. The authors state, for example, that in most cases barbed sutures enabled simple and rapid closure of defects and made knot tying unnecessary. If there is no comparison group, there is no evidence for such a statement. In my opinion, the inclusion of a control group would significantly improve the quality of this manuscript and is therefore necessary for publication!

5. Study Outcomes – The primary and secondary outcomes of the study are not mentioned in the methodology.

6. The authors state that after fully exposing and identifying the area of the defect, the defect is categorized according to size. Please add an adequate reference describing the categorization of the defect.

7. The authors state that Ethical approval for reporting the series was obtained from their institutional ethics board. Please indicate the approval number together with the date of approval.

8. Statistical analysis – Please specify the type of data processing. Even if only descriptive statistics were used, this should be stated along with information on how the data were expressed (e.g. median (IQR) or mean (SD) or n (%)). I hope that in the revised version, the authors will also include the comparison group and the statistical program/ tests used for comparison.

9. Tables – All tables are very reader-unfriendly. The variables studied should be presented in columns and each patient should be listed individually in rows. In that way, the Table will be much more reader-friendly. If a control group would be included all tables should be re-arranged.

10. In their discussion the authors state only one study dealing with barbed sutures for CDH repair in children. They should indicate that barbed sutures were used in similar pathology in adults (Wade A, et al. Hiatal hernia cruroplasty with a running barbed suture compared to interrupted suture repair. Am Surg. 2016;82(9):e271-4).

11. The retrospective design of the study should be listed under the limitations, as well.

12. The References list should be updated. As per MDPI standards, a minimum of 30 references is required for such type of study. In addition, references are not presented as per journal style. Please visit the instructions and update.

13. The quality of the English language should be improved. The study would benefit from professional editing.

Comments on the Quality of English Language

The quality of the English language should be improved. The study would benefit from professional editing.

Author Response

  1. Reviewer 1

The authors reported their initial experience of repairing congenital diaphragmatic hernias using barbed sutures, including the pitfalls they encountered. They concluded that barbed sutures simplify the repair of congenital diaphragmatic hernias, regardless of whether a minimally invasive or open approach is chosen. Patch repair is not a contraindication to the use of barbed sutures. The resulting potential time savings make it particularly useful in patients with cardiac or other severe comorbidities, in which shorter operative times are essential.

I read the study with great interest. Unfortunately, important methodological issues were raised during the review. My concerns are as follows:

1.1. Abstract – Authors should indicate the average age of patients in the abstract. In addition, any abbreviation in the abstract should be listed in the full title on the first mention (e.g. ECMO).

> Thank you very much. We have included the age of the patients and spelled out ECMO.

1.2. Introduction – The authors state that barbed sutures have been used for a variety of indications, including cosmetic plastic surgery, obstetrics and gynecology, gastrointestinal surgery, and orthopedic surgery. This is true, but barbed sutures were also used in pediatric urology (e.g. for pyeloplasty). This statement should be added including a recent meta-analysis reference (Anand S, et al. Barbed versus non-barbed suture for pyeloplasty via the minimally invasive approach: A systematic review and meta-analysis. J Laparoendosc Adv Surg Tech A. 2022;32(10):1056-1063. doi: 10.1089/lap.2021.0868).

> Thank you very much for pointing this out. We have included the reference to pediatric urology, pyeloplasty, and the reference mentioned [the following references were renumbered].

1.3. Study design – The authors state that demographics, operative parameters, complications, and outcomes were recorded in a registry and later analyzed. Please indicate for each category the analyzed parameters.

> The categories are marked in the respective tables (demographics in table 1, operative parameters in table 2, complications and outcomes in table 3).

1.4. Study Design – The main problem with the design of this study is the lack of a comparison group. I would advise the authors to introduce a comparison group (Patients operated on with other types of sutures from previous years). Without a comparison group, this is a descriptive study without clinical or scientific relevance. The authors state, for example, that in most cases barbed sutures enabled simple and rapid closure of defects and made knot tying unnecessary. If there is no comparison group, there is no evidence for such a statement. In my opinion, the inclusion of a control group would significantly improve the quality of this manuscript and is therefore necessary for publication!

> We included a historic control group (table 4). This is also discussed in a separate paragraph in the discussion section (see page 10, second paragraph). A box plot of operative times (figure 4) is also included.

1.5. Study Outcomes – The primary and secondary outcomes of the study are not mentioned in the methodology.

> Operative time was defined as the primary outcome measure. All other parameters were defined as secondary outcome parameters. In 2.1 Study design, a sentence was added referring to comparison to the historic control group: "The operative time as the primary outcome parameter was statistically compared to a historic control group using the t-test."

1.6. The authors state that after fully exposing and identifying the area of the defect, the defect is categorized according to size. Please add an adequate reference describing the categorization of the defect.

> Article on standardized reporting of congenital diaphragmatic hernia added as reference 7 in 2.4. Surgical Technique. The following references were renumbered.

1.7. The authors state that Ethical approval for reporting the series was obtained from their institutional ethics board. Please indicate the approval number together with the date of approval.

> Ethics board approval number (23-0881) included under 2.5. Consent and ethics.

1.8. Statistical analysis – Please specify the type of data processing. Even if only descriptive statistics were used, this should be stated along with information on how the data were expressed (e.g. median (IQR) or mean (SD) or n (%)). I hope that in the revised version, the authors will also include the comparison group and the statistical program/ tests used for comparison.

> Regarding the primary outcome, please see response 1.5. above. For secondary parameters, descriptive stastistics were employed, a sentence was added in 2.1 Study design: "All other parameters were defined as secondary measures, which were reported using de-scriptive statistics in terms of frequency, median and range."

1.9. Tables – All tables are very reader-unfriendly. The variables studied should be presented in columns and each patient should be listed individually in rows. In that way, the Table will be much more reader-friendly. If a control group would be included all tables should be re-arranged.

> The tables were all transposed and reformatted as requested

1.10. In their discussion the authors state only one study dealing with barbed sutures for CDH repair in children. They should indicate that barbed sutures were used in similar pathology in adults (Wade A, et al. Hiatal hernia cruroplasty with a running barbed suture compared to interrupted suture repair. Am Surg. 2016;82(9):e271-4).

>Although cruroplasty is a fundamentally different operation, we have included this refference in the second paragraph of the discussion.

1.11. The retrospective design of the study should be listed under the limitations, as well.

> Retrospective nature of the study was added as a limitation in the last paragraph of the discussion.

1.12. The References list should be updated. As per MDPI standards, a minimum of 30 references is required for such type of study. In addition, references are not presented as per journal style. Please visit the instructions and update.

>The references were updated and expanded to 18 references. This covers all aspects mentioned in the paper, there simply is not much published data on barbed sutures for CDH (except for 1 study, [7]. We do not believe that references should simply be added for the sake of it, or to fulfill a certain quota.

1.13. The quality of the English language should be improved. The study would benefit from professional editing.

> Both the first and last author are native English speakers. We have written the article to the best of our abilities. Please, could you give us specific examples where we can improve the language?

Reviewer 2 Report

Comments and Suggestions for Authors

The authors present their initial experience in using barbed sutures to close the diaphragmatic defect in children with congenital diaphragmatic hernia. Their results indicate that the use of barbed sutures is safe and effective and, by “obviating the need for knot-tying”, has the potential to reduce the operative time. However, the study lacks the control group and the group is relatively small. More data and larger studies are need to draw a conclusion. The main merit of the report is the idea of using this type of sutures for diaphragmatic hernia. Overall, the manuscript is well written and easy to follow.

There are a few issues:

Line 83 – The severity grading of the defect requires proper citation

Line 84 – Please mention that barbed sutures don/t have to be tied up. What type of suturing was used: running or isolated?

Line 215 - Cost of the barbed sutures. Data regarding the costs of the sutures shall be first presented in the methodology or result sections, with proper citation if possible and only afterwards discussed

The references are not in the style of the journal

Author Response

  1. Reviewer 2

The authors present their initial experience in using barbed sutures to close the diaphragmatic defect in children with congenital diaphragmatic hernia. Their results indicate that the use of barbed sutures is safe and effective and, by “obviating the need for knot-tying”, has the potential to reduce the operative time. However, the study lacks the control group and the group is relatively small. More data and larger studies are need to draw a conclusion. The main merit of the report is the idea of using this type of sutures for diaphragmatic hernia. Overall, the manuscript is well written and easy to follow.

There are a few issues:

2.1. Line 83 – The severity grading of the defect requires proper citation

> See answer to comment 1.6. A reference on standardized reporting of diaphragmatic hernia was added.

2.2. Line 84 – Please mention that barbed sutures don/t have to be tied up. What type of suturing was used: running or isolated?

> Additional sentence added to the 3rd paragraph of the introduction

2.3. Line 215 - Cost of the barbed sutures. Data regarding the costs of the sutures shall be first presented in the methodology or result sections, with proper citation if possible and only afterwards discussed

> The issue of cost was expanded and references given

2.4. The references are not in the style of the journal

> The references were reformatted.

Reviewer 3 Report

Comments and Suggestions for Authors

Very interesting and important study for a pediatric surgeon. The only thing that attracts attention is table no. 3, which is not very legible and clear, and a small number of references.

Author Response

  1. Reviewer 3

Very interesting and important study for a pediatric surgeon. The only thing that attracts attention is table no. 3, which is not very legible and clear, and a small number of references.

> Thank you very much for your comments. The tables were transposed for clarity, see response 1.9. above. Additional references were added.

Reviewer 4 Report

Comments and Suggestions for Authors

Dear Sirs

According to the title, this was supposed to be a very straightforward study addressing the use of barbed sutures in the management of CDH repair. Although the subject is interesting, the advantages of barbed sutures have already been described and the study does not bring any new information to what is already known to the subject. The lack of a control group, although justified by the authors, precludes any definitive conclusion about any advantages of the used of this type of sutures over conventional ones. Additionally, it is believed that, in the result sections, if the stady is about the advantages of the use of barbed sutures, there is no need to present all the epidemiological and clinical data about the patients and their follow up. This resulted in a very long and hard to read table that does not add any significant information to this manuscript.

In summary, although addressing an interesting subject, due to a inadequate experimental design, this  manuscript fails to add any significant information that could help in the clinical management of CDH

Author Response

  1. Reviewer 4

According to the title, this was supposed to be a very straightforward study addressing the use of barbed sutures in the management of CDH repair. Although the subject is interesting, the advantages of barbed sutures have already been described and the study does not bring any new information to what is already known to the subject. The lack of a control group, although justified by the authors, precludes any definitive conclusion about any advantages of the used of this type of sutures over conventional ones. Additionally, it is believed that, in the result sections, if the stady is about the advantages of the use of barbed sutures, there is no need to present all the epidemiological and clinical data about the patients and their follow up. This resulted in a very long and hard to read table that does not add any significant information to this manuscript.

In summary, although addressing an interesting subject, due to a inadequate experimental design, this  manuscript fails to add any significant information that could help in the clinical management of CDH

> In order to overcome this obvious limitation, a historic control group was added. We do feel that the shortened operative time shown in this study helps in the clinical management of CDH, since shorter operative times are particularly important in children with co-morbidities.

Round 2

Reviewer 1 Report

Comments and Suggestions for Authors

The authors have improved the manuscript considerably, in particular by adding a comparison group.

I have two suggestions before acceptance:

-        Tables 1-3 should be merged into one table (as they presented a comparison group).

-        A comparative table with the median values of all investigated variables should be added. A statistical comparison of all variables studied should also be performed.

Comments on the Quality of English Language

Minor editing of English language required

Author Response

Dear Prof. Carney, dear CHILDREN editorial board,

thank you very much again for allowing us to improve our manuscript. The changes according to the comments of the reviewers are detailed below and marked in blue in the text.

We hope that these changes make the article acceptable for publication in CHILDREN.

Please don't hesitate to contact us for any further questions or concerns.

Best regards,

Nadine Muensterer and co-authors

___________________________________________________________________________

Reviewer 1:

The authors have improved the manuscript considerably, in particular by adding a comparison group. I have two suggestions before acceptance:

- Tables 1-3 should be merged into one table (as they presented a comparison group).

>The tables were merged into one table (table 1), the control group was presented in table 2.

- A comparative table with the median values of all investigated variables should be added. A statistical comparison of all variables studied should also be performed.

> The statistical comparison of all variables was added at the bottom of table 2, rather than in a separate table

Reviewer 4:

The manuscript was improved by the addition of a historic control series and several other corrections. My only suggestion is that, if the lack of significance on the operative time is due to the outliers, you should exclude the outliers from the analysis (and explain why you did that in the text). The two groups should be matched and comparable in terms of type and size of the defect and type of surgery performed. Otherwise comparisons do not mean anything.

> We have recalculated the p-values with and without the outliers and have discussed this in the text.

Besides this, I believe that the manuscript has been greatly improved.

Reviewer 4 Report

Comments and Suggestions for Authors

Dear Sirs

The manuscript was improved by the addition of a historic control series and several other corrections. My only suggestion is that, if the lack of significance on the operative time is due to the outliers, you should exclude the outliers from the analysis (and explain why you did that in the text). The two groups should be matched and comparable in terms of type and size of the defect and type of surgery performed. Otherwise comparisons do not mean anything.

Besides this, I believe that the manuscript has been greatly improved.

Author Response

(The authors gave the same response as above.)
